# Recombinant *Helicobacter pylori* Vaccine Delivery Vehicle: A Promising Tool to Treat Infections and Combat Antimicrobial Resistance

**DOI:** 10.3390/antibiotics11121701

**Published:** 2022-11-25

**Authors:** Yakhya Dieye, Cheikh Momar Nguer, Fatou Thiam, Abou Abdallah Malick Diouara, Cheikh Fall

**Affiliations:** 1Groupe de Recherche Biotechnologies Appliquées & Bioprocédés Environnementaux (GRBA-BE), École Supérieure Polytechnique, Université Cheikh Anta Diop, Dakar BP 5085, Senegal; 2Pôle de Microbiologie, Institut Pasteur de Dakar, 36 Avenue Pasteur, Dakar BP 220, Senegal

**Keywords:** antimicrobial resistance, oral vaccination, recombinant vaccine vehicle, *Helicobacter pylori*

## Abstract

Antimicrobial resistance (AMR) has become a global public health threat. Experts agree that unless proper actions are taken, the number of deaths due to AMR will increase. Many strategies are being pursued to tackle AMR, one of the most important being the development of efficient vaccines. Similar to other bacterial pathogens, AMR in *Helicobacter pylori* (*Hp*) is rising worldwide. *Hp* infects half of the human population and its prevalence ranges from <10% in developed countries to up to 90% in low-income countries. Currently, there is no vaccine available for *Hp*. This review provides a brief summary of the use of antibiotic-based treatment for *Hp* infection and its related AMR problems together with a brief description of the status of vaccine development for *Hp*. It is mainly dedicated to genetic tools and strategies that can be used to develop an oral recombinant *Hp* vaccine delivery platform that is (i) completely attenuated, (ii) can survive, synthesize in situ and deliver antigens, DNA vaccines, and adjuvants to antigen-presenting cells at the gastric mucosa, and (iii) possibly activate desired compartments of the gut-associated mucosal immune system. Recombinant *Hp* vaccine delivery vehicles can be used for therapeutic or prophylactic vaccination for *Hp* and other microbial pathogens.

## 1. Introduction

The discovery of penicillin, the first antibiotic, by Sir Alexander Fleming represented a major breakthrough in treating infectious diseases. This initial discovery was followed by many other natural or synthetic molecules, and antibio-therapy has since saved millions of lives worldwide. However, as anticipated by Fleming himself at the time of his discovery, bacteria have developed abilities to resist antibiotics at a rate similar to or higher than the availability of new molecules. Today, AMR has become a global public health threat. Multidrug resistant (MDR) clones of important human pathogens are emerging at an alarming rate [1], and experts agree that if proper actions are not taken, the number of deaths due to resistant bacteria will increase in the future. The increasing trend of AMR was confirmed by a recent review showing the global burden of AMR for 2019 [2]. Not surprisingly, sub-Saharan Africa (sSA) was the world region with the highest death rates directly due to or associated with AMR [2]. To address AMR, many strategies are being conducted both locally and globally. These strategies are based on (i) the production and sharing of quality data on AMR, (ii) capacity building for bacterial culture, antimicrobial susceptibility testing and its interpretation, (iii) surveillance based on a one health approach and (iv) development of alternatives to antibiotic-based treatment. Although challenging, the development of prophylactic and therapeutic vaccines appears to be a powerful means that can be harnessed to tackle the rise of AMR. This strategy is strongly advocated by international organizations at the forefront of the efforts developed to tackle the rise of AMR including the WHO, FAO, OIE. The rapid and efficient development of different vaccines against SARS-CoV-2 during the COVID pandemic has shown that existing scientific tools can serve this purpose.

Similar to other bacterial pathogens, AMR in *Helicobacter pylori* (*Hp*) has become an important public health concern, given the increasing prevalence of clones resistant to antibiotics used to treat infection [3]. This is especially true in sSA, where the prevalence of *Hp* is high and quality data on the resistance of circulating clones are scarce. Alternative methods to manage *Hp* infection and its associated diseases are most needed in sSA and other low-income countries (LIC). Development of therapeutic and prophylactic vaccines against *Hp* is, without doubt, the best means to cure and prevent infection respectively, and reduce the burden of diseases associated with this bacterium. While there is no available vaccine against *Hp*, this short review intends to present pieces of evidence that recombinant *Hp* can be developed to serve as an oral vaccine delivery platform. After short summaries on AMR and the status of vaccine development for *Hp*, this review will discuss detailed knowledge of the available tools that can be used to construct recombinant *Hp* vehicles that are safe, can synthesize and deliver antigens in situ, and activate different compartments of the gut-associated mucosal immune system (G-MIS). *Hp* delivery platform can be used not only for *Hp*, but for other microbial pathogens as well.

## 2. Treatment of *Helicobacter pylori* Infection and Problems Posed by Eradication Failure

*Hp* is one of the most successful bacterial colonizers of humans. It has been associated with mankind for over 100,000 years and today, the average prevalence of *Hp* infection globally is >50% [2]. This prevalence varies greatly, ranging from <10% to >90%, with large differences between countries and between communities in the same country [2]. Several factors contribute to *Hp* prevalence and its associated diseases, including socioeconomic conditions, host genetic predisposition, age, ethnicity, and environmental factors [2,3]. Poverty is one of the main contributors to the *Hp* burden and explains the consistent differences between developed and developing countries. *Hp* infection often lasts for the host’s lifetime if untreated. In most individuals, chronic infection by *Hp* results in asymptomatic stomach inflammation. In subsets of subjects, *Hp* carriage leads to different diseases, including peptic ulcers, chronic atrophic gastritis, gastric cancers, and gut lymphoma [4,5].

Treatment of *Hp* infection is based on the use of a combination of drugs, including one to three antibiotics in addition to a proton pump inhibitor (PPI) and/or bismuth salts [6]. PPIs inhibit gastric acid secretion [7], while bismuth has a direct killing effect against *Hp* [8]. Both PPIs and bismuth act synergistically with antibiotics against *Hp* [9,10]. The antibiotics used in the *Hp* eradication regimen include clarithromycin, metronidazole, ampicillin, fluoroquinolone, and tetracycline [11]. Nowadays, recommendations for first-line *Hp* treatment depend on local rates of resistance to the above-mentioned antibiotics and may correspond to PPI-based triple therapy (PPI plus two antibiotics), bismuth-based quadruple therapy (bismuth, PPI, and two antibiotics), non-bismuth concomitant quadruple therapy (PPI plus three antibiotics) or other combinations [12]. In case of treatment failure, it is advised to perform *Hp* isolation and antibiotic susceptibility testing (AST) before using a different eradication regimen [13]. This requires biopsies to be taken and *Hp* isolated. In LIC, especially in sSA, there is a lack of qualified microbiologists and of laboratories properly equipped to perform *Hp* culture and AST. Consequently, despite high carriage, reports of *Hp* prevalence and incidence in the continent greatly vary among regions, with certain countries making available quality data while there are rare to no data for the majority. Therefore, the resistance profile of circulating clones is mostly unknown and treatment is largely empirical. An efficient vaccine is, without doubt, the best means to reduce *Hp* infection and its associated diseases in sSA.

## 3. Current Status of Vaccines for *Helicobacter pylori*

There is currently no vaccine approved for *Hp*. Despite many candidates developed in preclinical studies (Table 1), *Hp* vaccines are poorly performed in clinical trials [11]. This is primarily due to a fascinating arsenal of immune escape strategies that *Hp* has developed during its long co-evolution with humans [14,15]. *Hp* immune escape targets both innate [16,17,18] and adaptive immune systems [17,18,19,20] leading to unresponsiveness or to activation of inappropriate responses that fail to sterilize the host. Nevertheless, recent advances in understanding *Hp* pathogenesis and its interaction with the host immune system, in addition to the availability of genetic engineering tools enable a better design of potentially successful vaccines. Important *Hp* virulence factors including urease, flagellar subunit, catalase, CagA, VacA, NapA, HpaA, immune escape proteins OipA and GGT, adhesion BabA, SabA, and Omp, have been well-studied [19,21]. Innovative approaches, such as immunoinformatics, have been used to identify various epitopes that can be combined in multiepitope vaccines [22] including for oral delivery [23]. These novel approaches can be combined in a recombinant *Hp* vaccine platform that can immunize for *Hp* and other pathogenic microbes.

## 4. *Helicobacter pylori* as a Platform for the Delivery of Oral Vaccines

As one of the most successful bacterial colonizers of humans, *Hp* can be an ideal vehicle for delivering biologically active molecules, including vaccines and therapeutic drugs, to the digestive tract [35]. *Hp* presents several advantages for this type of application. Firstly, there are avirulent (type II) strains [36] and the most important virulence factors of *Hp* have been well characterized and can be easily inactivated. Secondly, *Hp* efficiently colonizes the human stomach, a site that is less populated by microbial flora than the intestines. Therefore, recombinant *Hp* cells expressing vaccines will face less competition for the occupation of gastric niches, a factor that impacts the delivery of antigens. Thirdly, *Hp* can gain access to gastric lymph nodes [37,38] where it encounters antigen-presenting cells (APC) and other immune cells. Additionally, *Hp* cells can reach Peyer’s patches in the small intestine, yet another important immune induction site [39]. Therefore, *Hp* can deliver antigens to two primary G-MIS induction sites. Fourthly, substantial knowledge has been accumulated on the physiology and the pathogenicity of *Hp*. Several aspects of the interaction of the bacterium with the G-MIS are known at the molecular level [40]. This makes it possible to selectively inactivate genes involved in immune escape or to construct recombinant strains that favor the desired type of immune response. Fifthly, it is possible to take advantage of the technologies, approaches, and strategies developed in other models of bacterial vaccine delivery vehicles including attenuated pathogen and commensal/non-pathogenic vectors.

### 4.1. Genetic Tools and Technologies for Recombinant Helicobacter pylori Vaccine Delivery Vehicle

An intermediary step toward the use of *Hp* as a vaccine delivery vehicle is the design of genetic and molecular tools that enable modification of the genome, expression, export, and targeting of antigens in different compartments of the bacterium. *Hp* is a fastidious bacterium that has numerous requirements for growth. Also, there are fewer tools for genetic manipulation available in *Hp* than in model organisms such as *E. coli* and *Salmonella*. Early mutagenesis systems developed in *Hp* included a few random and targeted mutagenesis strategies using transposon and suicide-plasmid based systems [41,42]. The development of a mutagenesis system based on a counterselectable marker represented a significant advance in *Hp* mutagenesis [43]. In this system a two-gene DNA cassette specifying an antibiotic resistance marker and a dominant allele of the *rpsL* ribosomal gene that confers sensitivity to streptomycin is used to replace chromosomal gene. Mutations constructed with this system are easily transferable to other *Hp* strains by natural transformation using genomic DNA from the mutant. Unmarked deletions can be efficiently obtained after the excision of the mutating cassette from the chromosome. Following this, Xer-Cise targeted mutagenesis that enables one-step construction of unmarked chromosomal deletions has been developed in *Hp* [44]. Xer-Cise uses the site-specific recombinase XerH, and its cognate target *dif* sequence that is involved in the resolution of chromosome dimers [45]. In this system, the two-gene counterselectable cassette described above is flanked by *dif* sequences and used to replace the chromosomal genes (Figure 1). Subsequently, unmarked deletions are counterselected without the need for the introduction of additional DNA material. XerCise system is very similar to the lambda red and Frt-Flip technologies first developed in *E. coli* [46] and as with that method could possibly be used to insert vaccine constructs in *Hp* chromosome [47] (Figure 1).

### 4.2. Recombinant Protein Expression Systems for Helicobacter pylori

The development of recombinant vaccine delivery vehicles requires expression systems that enable sufficient and stable production of antigens. Both chromosomes and plasmids have been used as locations for antigen constructs. Chromosomal expression presents the advantage of the stability of vaccine constructs. However, it suffers from low levels of protein production. In contrast, plasmids that can exist in multiple copies in a bacterial host often enable high expression levels. Several plasmid vectors have been developed in *Hp* and used to express foreign genes [48]. However, plasmid expression suffers a few drawbacks for use in *Hp* delivery vehicles. Firstly, since *Hp* can be difficult to manipulate, vaccine constructs would typically be performed in easily manipulable hosts such as *E. coli* before transfer to *Hp*. Although natural competence and conjugation systems enable the introduction of DNA material into *Hp* cells, the success rate is often low and varies greatly from strain to strain. This low success is likely due to the strong restriction enzyme barriers that exist in *Hp* [49]. Secondly, constructs expressed from plasmids are more susceptible to instability and additionally can negatively affect the fitness of the bacterial vehicle during host colonization because of the metabolic burden that results from plasmid maintenance and gene expression. Several alternatives can be proposed to enable stable expression of vaccine constructs in *Hp* without affecting bacterial fitness in host tissues. The design of constructs with optimized translational signals and their insertion in the chromosome as transcriptional fusions with highly expressed genes may support a sufficient level of antigen production. For example, urease, constituted by UreA and UreB subunits, is highly expressed making up to 10% of total *Hp* proteins [50]. The regulation of the *ureAB* operon has been well studied and this enzyme is essential for the colonization of gastric tissues [51]. Transcriptional fusion of vaccine constructs with *ureAB* operon can be a means to achieve in vivo induction of antigen production. Antigen expression can be further enhanced by optimizing translational signals (ribosome binding site and sequence around the initiation codon) of vaccine constructs. Another alternative could be the use of inducible promoters on the middle to high copy-number plasmids. Unfortunately, only a few inducible systems are available in *Hp* [52] and their functionality during colonization of the host’s stomach was not evaluated. Interestingly, in vivo-induced promoters have been identified in *Hp* cells infecting mice [53]. These promoters could be used to design vaccine constructs that are repressed or moderately expressed (from chromosome or plasmid) during growth in vitro and yet highly expressed in the host stomach. Such an approach can relieve the metabolic burden that results from the constitutive expression of heterologous vaccine constructs and improve in situ synthesis and delivery of antigens.

### 4.3. Selection of Recombinant Bacteria and Stabilization of Antigen-Encoding Plasmids

The use of a plasmid for antigen expression poses two problems that need to be addressed in a vaccine delivery vehicle. Firstly, commonly used plasmids harbor antibiotic resistance genes that enable the selection of recombinant bacteria. However, the use of antibiotic markers in vaccine carriers is unacceptable for regulatory reasons. Indeed, resistance gene-encoding plasmids can be horizontally transferred to other bacteria that reside in the same niches as the recombinant vehicle during host colonization. The risk of dissemination of drug resistance to other bacteria, including pathogens, poses threats that make necessary the prohibition of antibiotic resistance markers from recombinant bacteria destined to be used in human or animal health. The use of antibiotic markers has been addressed by developing antibiotic-free plasmids that can be used for biotechnology purposes [54,55,56,57]. These antibiotic marker-free plasmids are based on different mechanisms including complementation of auxotrophic mutation [40,58], operator-repressor titration [43,44], plasmid addiction such as the toxin-antitoxin system [45], RNA-based regulation [59,60] and resistance to a non-antibiotic marker [61]. Besides precluding the use of antibiotic selection, the mechanisms mentioned above also contribute to stabilizing recombinant plasmids enabling the continuous expression of vaccine constructs. Indeed, loss of the plasmid results in the death of the bacterial cells. Plasmid instability and loss are frequent in recombinant bacteria during host colonization, especially for high copy number replicons. This is due to the metabolic burden resulting from plasmid maintenance and gene expression. Plasmid loss or instability in the bacterial vaccine carrier compromises the expression and delivery of antigens, and ultimately immune response. Furthermore, inducible expression systems have been incorporated in these vectors in order to provide a controlled expression of vaccine constructs and find an optimal balance between antigen production and bacterial fitness in host organs [62,63,64].

The antibiotic-free vectors have been widely in the two existing categories of bacterial vaccine delivery vehicles including attenuated pathogens and non-pathogenic/commensal bacteria, the most represented species of the two types being *Salmonella* [64] and *Lactococcus lactis* [63]. We describe below the balanced-lethal vector-host system developed in *Salmonella*, a vehicle of the same category as *H. pylori*, as an example of one of the most attracting systems that serve as both antibiotic-free and stabilization systems and additionally enable within-host inducible expression of vaccine construct. Balanced-lethal system relies on a gene that is (i) essential and (ii) can be complemented by the addition of a metabolite in growth media. Bacterial vaccine carriers lack the essential gene and require supplementation of the metabolite for growth. The essential gene is cloned in an expression plasmid that when introduced in the vaccine carrier provides complementation. Growth media not supplemented with the metabolite are used to select for bacterial cells carrying the complementing plasmid. A balanced-lethal system developed in *Salmonella* used the *asdA* gene that encodes aspartate β-semialdehyde dehydrogenase, an enzyme required for the synthesis of diamino pimelic acid (DAP). DAP is a unique constituent of the peptidoglycan layer of bacterial cell wall necessary to maintain cell shape and stability. DAP is only synthesized by bacteria and is unavailable in eukaryotic tissues. The DAP synthesis pathway has been studied in *Hp* [65] and a requirement for DAP was demonstrated in *Hp dapE* mutants. Therefore, this system can be implemented in *Hp* to construct an antigen expression vector (Figure 2). An important improvement of the *asd* balance-lethal system of *Salmonella* consisted in adding a plasmid encoding a regulated repressor of the promoter governing the expression of the vaccine construct enabling the induction of antigen production in host organs [64]. This delayed regulated expression improves the fitness of the recombinant delivery vehicle in the host.

### 4.4. Targeting of Antigen Constructs in Helicobacter pylori

The subcellular location of an antigen in a bacterial carrier may impact the immune response [66,67]. It is therefore important to test antigens expressed in different compartments of the *Hp* delivery vehicle including cytoplasm, cell surface, or release into the surrounding environment (Figure 3). Cytoplasmic expression is straightforward and mainly requires optimal transcription and translation signals. Export of antigens outside *Hp* cells might be more challenging. Several export systems that enable protein crossing of the inner and outer membrane exist in Gram-negative bacteria [68]. Several of these systems have been characterized in or identified in the genome of *Hp*. Sec-dependant, twin-arginine translocation, lipoprotein, type IV secretion, and autotransporter pathways are present and studies of the *Hp* secretome have identified a number of proteins that are released in the supernatant of liquid cultures [69,70,71]. In addition, the *Hp* genome encodes dozens of outer membrane proteins (OMP) that are targeted to the bacterial surface by different mechanisms [72]. Export and targeting signals from these exported proteins can be used to construct recombinant *Hp* cells that display antigens at the cell surface or release them into the surrounding environment. Similarly, the Cag (cytotoxin-associated gene) type four secretion system (T4SS) could be engineered to construct *Hp* cells that directly inject antigens inside eukaryotic cells, a location particularly efficient for induction of a cell-mediated immune response [73,74]. Another means of delivery could be through outer membrane vesicles (OMV). OMVs are naturally produced by Gram-negative bacteria and OMV biogenesis is a controlled and regulated process that fulfills several functions [75]. *Hp* OMVs have been shown to interact with epithelial cells and to deliver bacterial proteins to host cells [76,77]. Although the mechanism of OMV biogenesis and regulation in Gram-negative bacteria is not fully elucidated, it can be anticipated that fusion to proteins located in *Hp* OMVs [77,78] could be a means of antigen delivery to eukaryotic cells.

### 4.5. Delivery of DNA Vaccines by Helicobacter pylori

A potential limitation of antigens delivered by bacteria is the absence of post-translational protein modifications that occur in eukaryotic cells but not in bacteria. These modifications can lead to the formation of epitopes that induce protective immunity against viruses and eukaryotic microbes. Delivery of antigen-encoding DNA to host cells represents a means to overcome such a limitation. Besides this, DNA vaccines present other advantages. Firstly, DNA eliminates the need for antigen synthesis that imposes a metabolic burden on the bacterial vaccine carrier and negatively affects its fitness in host tissues. Secondly, antigens synthesized in the cytosol of host cells are efficient in inducing cell-mediated immunity, are properly processed by host chaperones, and preserve both linear and conformational epitopes. Thirdly, DNA molecules can be engineered to harbor innate immune activators such as CpG motifs that are recognized by TLR (toll-like receptor) 9 [79]. Several attenuated bacterial pathogens have been used to deliver DNA vaccines including *Salmonella*, *Shigella,* and *Vibrio* [80]. In these systems, antigen-encoding DNAs are harbored by plasmids that additionally contain signals for efficient antigen synthesis by eukaryotic cells including nuclear transport, eukaryotic transcriptional, and translational signals. *Hp* presents a potential for the delivery of antigen-encoding DNA to eukaryotic cells. DNA transfer between *Hp* cells during growth in vitro and in host tissues is established [81,82,83]. Plasmid transfer by *Hp* can occur through at least three pathways including natural transformation, conjugation via T4SS, and a recently identified yet not fully characterized mechanism [84]. Interestingly, Cag-T4SS can deliver chromosomal *Hp* DNA into host cells [85,86]. To be effective in inducing an immune response, a DNA vaccine should be delivered to APC. *Hp* interacts, is internalized, and survives in gastric APC including dendritic cells [87] and macrophages [88]. These findings indicate that recombinant *Hp* could possibly be engineered to deliver DNA vaccine constructs into gastric APC. 

### 4.6. Overcoming Oral Tolerance and Immune Suppression by Helicobacter pylori

The design of a *Hp* oral vaccine delivery system needs to take into account oral tolerance, a state of immune unresponsiveness to an antigen delivered by the oral route [89] on the one hand, and the diverse immune escape strategies used by *Hp* on the other. These two issues can be properly addressed by using as a vaccine delivery vehicle, strains with a mutation in one or more immune escape strategies and that additionally express one or several adjuvants. Mucosal tolerance is an essential regulatory mechanism of the G-MIS that prevents inappropriate and detrimental activation against non-threatening compounds in the mucosa. Mucosal vaccine formulation typically includes mucosal adjuvants to overcome immune tolerance [90]. An adjuvant is an immune stimulant that, when co-administered with an antigen, enhances the potency and the quality of the immune response induced. The most efficient mucosal adjuvants comprise bacterial enterotoxins, TLR ligands, and cytokines. Bacterial enterotoxins, in particular ADP-ribosylating toxins such as cholera toxin (CT) and *E. coli* heat-labile toxin (LT), represent a class of mucosal adjuvants that have been well characterized [91]. These toxins are oligomeric proteins composed of one enzymatically active subunit and a homopentameric receptor binding subunit [92]. The toxin binds ganglioside GM1 [93] present at the surface of most eukaryotic cells including enterocytes and APC. The toxin’s binding to its receptor and its subsequent internalization activate signaling pathways that result in pleiotropic effects with activation of both innate and adaptive mucosal immune systems [94,95,96]. Concerns regarding the toxicity of the toxins have been solved by constructing mutants of CT and LT that are devoid of toxicity while retaining full adjuvant properties [95,97,98,99,100]. Additionally, it has been shown that the receptor-binding pentamer without the catalytic subunit still displays immunostimulatory activities [95,101,102,103]. For a *Hp* vaccine carrier, it is important to stress that CT and LT are bacterial proteins, and therefore can be efficiently synthesized and properly folded in a bacterial host. Another class of adjuvant used in mucosal immunization groups a number of TLR agonists [104]. TLR activation stimulates signaling pathways that result in the production of pro-inflammatory cytokines, recruitment of immune cells including APC, helper, and effector cells, and initiation of an adaptive immune response [104]. For a *Hp* vaccine delivery platform, the use of TLR signaling may primarily consist in restoring TLR4 and TLR5 signaling that are compromised by wild-type *Hp* cells [105]. In addition to these, recombinant *Hp* may be genetically engineered to express ligands of other TLRs. Cytokines represent a third class of adjuvant that can be considered for the *Hp* vaccine carrier. Co-delivery of antigen with cytokines has been tested with other bacterial vaccine carriers including attenuated pathogens [106] and commensal bacteria [63,66] resulting in enhanced immune responses against the antigen. The interest in co-delivery of cytokine with antigen lies in that this class of adjuvant represents an efficient means of directing immune activation toward a desired type of response. However, due to their pleiotropic effects cytokine expression by vaccine carriers should be carefully controlled to avoid adverse reactions in the host.

### 4.7. Bio-Containment of Recombinant Helicobacter pylori Vaccine Carrier

Bio-containment is an important concern when using genetically modified microorganisms [107]. Recombinant bacterial delivery vehicles should (i) be safe, (ii) lack or have limited abilities to uptake from or transmit genetic material to other microorganisms and (iii) be unable to disseminate once excreted outside their hosts. An ideal approach should combine not only one, but several bio-containment methods, providing redundancy such that failure of all the containment mechanisms becomes highly unlikely. Safety concern for a *Hp* vaccine carrier has been discussed above. Regarding the transmission of genetic materials, *Hp* is one of the most genetically diverse bacterial species likely because of efficient DNA import and export abilities along with a strong recombination system [108]. *Hp* strains can encode up to four types four secretion systems (TFSS) three of which are involved in DNA exchange including ComB-TFSS that mediate import from the environment, and TFS3 and TFS4 that are involved in conjugative transfer of chromosomal and plasmid DNA [108]. Additionally, a conjugation-independent mechanism not well characterized yet has been reported in *Hp* [84]. These mechanisms favor intra-species DNA exchange. Besides, the transmission of genetic material by *Hp* to other species can occur [109] although this is not well-documented. Acquisition of foreign DNA by recombinant *Hp* vehicle should be avoided in order to preserve its vaccine attributes. This can be accomplished by inactivating genes involved in DNA import. Interestingly, mutation of the *comB* gene prevented DNA uptake by natural transformation or by conjugation [84]. This gene is therefore a good candidate to be mutated to reduce foreign DNA uptake by *Hp* delivery vehicles. DNA export can be reduced by mutating relaxase genes of *Hp tfs3* and *tfs4* and by using expression plasmids devoid of any signal for mobilization in their sequences. Additionally, a plasmid addiction system can be used to design recombinant plasmids with reduced transferability. For example, a *Hp* vaccine carrier that harbors a chromosomal gene encoding an antitoxin can be used as a host of a plasmid expressing the cognate toxin. Transmission of the plasmid to other bacteria will result in the death of recipient microorganisms that normally do not encode antitoxin. Dissemination of recombinant *Hp* excreted by its host in the environment will presumably be limited since this bacterium mostly converts to coccoid cells that are a viable but non-culturable form outside the human stomach and its natural ecological niche [110]. However, additional strategies developed in other bacterial delivery vehicles including attenuated pathogens [111] and non-pathogenic vectors [107,111,112] can be imported to *Hp*. An example relevant to *Hp* is an elegant regulated delayed lysis in vivo developed in *Salmonella* in which the addition of arabinose during culture in vitro of the recombinant delivery vehicle induces the synthesis of the cell wall. After administration to the host, cell wall integrity is maintained in vivo during a few generations when the vaccine constructed is expressed and delivered after which the bacterial cells undergo lysis due to the absence of arabinose [111]. *Hp* harbors homologs of the *asdA* and *murA* genes involved in cell wall synthesis in *Salmonella* opening the possibility of using this system in this bacterium.

## 5. Conclusions

Vaccination is a strategy of choice to address the growing threat of AMR globally. The development of vaccines against the most critical human and animal bacterial pathogens will result in a decrease in antimicrobial consumption and of resistance to the antibiotics used for the treatment of infections. Active researches aiming to develop prophylactic and therapeutic vaccines, including orally delivered formulations are ongoing. *Hp* by its long association with humans, its ability to colonize stomach mucosa, and the health benefits it could confer could be an ideal oral vaccine delivery system. Little work has been dedicated to this topic so far. The extensive knowledge accumulated on the pathogenesis of *Hp*, especially on its immune escape mechanisms, and the available genetic engineering tools enable the construction of recombinant strains that are completely attenuated while retaining the ability to colonize and multiply in the gastric mucosa. These strains can be vectored to carry multiple antigens or DNA vaccines that are synthesized in situ and delivered with immune stimuli and immune modulation molecules enabling an efficient activation of all the desired compartments of the immune system including the innate, humoral, cell-mediated, and memory branches. Furthermore, recombinant *Hp* delivery vectors can be endowed with biocontainment attributes precluding their dissemination outside their host. These oral vaccines will represent an important breakthrough that will contribute to addressing AMR in *Hp*. Besides *Hp*, they can serve as a delivery vehicle for oral vaccines against other microbial pathogens most relevant for AMR. Oral *Hp* vaccine will be especially suited for LIC such as in sSA where the threat of AMR is most pressing.

## Figures and Tables

**Figure 1 antibiotics-11-01701-f001:**
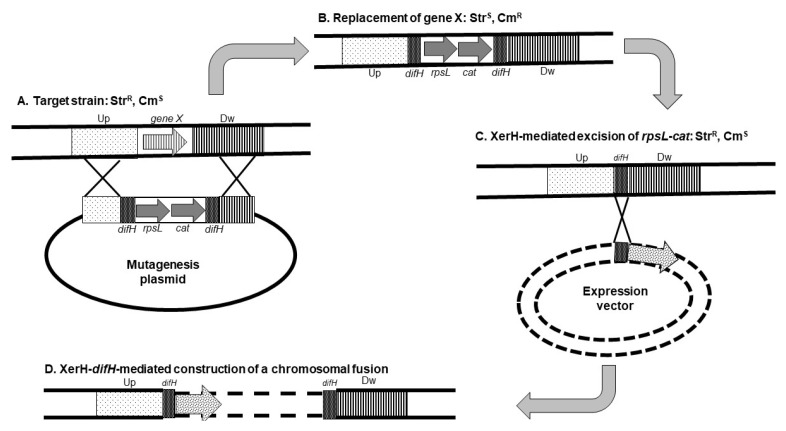
Construction of unmarked deletion and chromosomal fusion in *Helicobacter pylori*. A mutagenesis plasmid is introduced in the strain from which a gene (gene X) is to be deleted. This strain harbors a mutation of the *rpsL* gene that confers resistance to streptomycin (Str^R^). The mutagenesis plasmid contains the following elements: (i) two copies of *difH,* a target of the site-specific recombinase XerH flanking (ii) an *rpsL* allele that restores sensitivity to streptomycin (Str^S^), and (iii) a *cat* gene that confers resistance to chloramphenicol (Cm^R^); the *difH-rpsL-cat-difH* cassette is flanked by (iv) regions corresponding to sequences upstream (Up) and downstream (Dw) of the gene to be deleted. After the introduction of the mutagenesis plasmid (**A**), the target gene is replaced by the *difH-rpsL-cat-difH* cassette via homologous recombination between the upstream and downstream sequences from the plasmid and the chromosome; the resulting strain has an Str^S^-Cm^R^ phenotype (**B**). Growth of this strain in the absence of chloramphenicol and the presence of streptomycin enables the selection of excision of the *rpsL-cat* cassette via the activity of *H. pylori* XerH recombinase (**C**) leaving a single *difH* copy at the chromosomal location of the deleted gene (Str^R^-Cm^S^ phenotype). This *difH* copy can be used to insert a genetic construct in the chromosome of the resulting strain (**D**). Partially adapted from [30].

**Figure 2 antibiotics-11-01701-f002:**
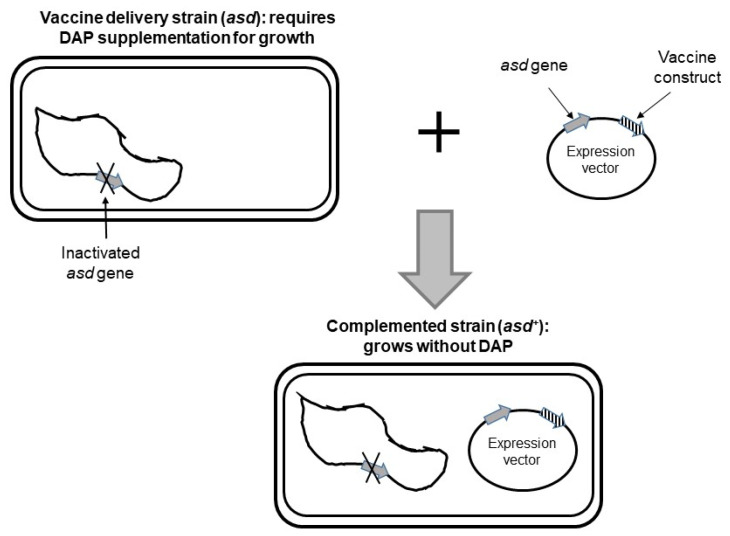
Balanced-lethal expression system for *Helicobacter pylori*. The recombinant vaccine carrier strain lacks *asd* gene and requires supplementation of diaminopimelic acid (DAP) for growth in vitro. Introduction of a plasmid vector harboring *asd* gene and expressing a vaccine construct complements for inactivated chromosomal *asd* gene. Adapted from [40].

**Figure 3 antibiotics-11-01701-f003:**
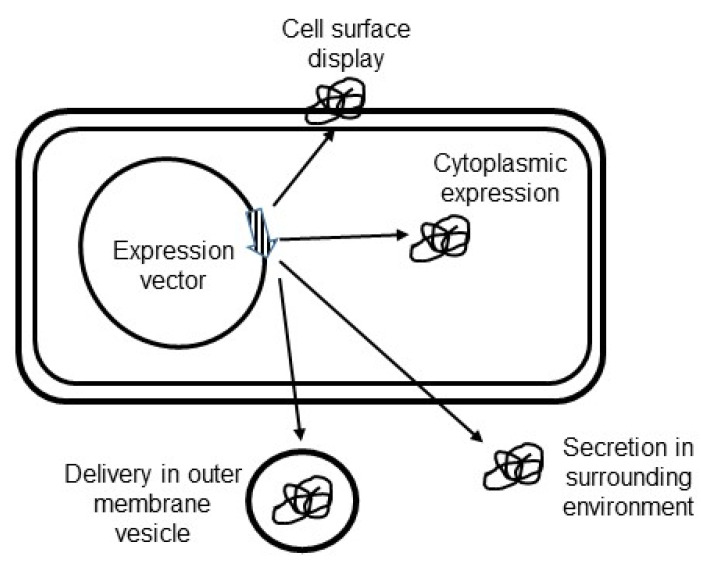
Antigen targeting in *Helicobacter pylori* vaccine delivery vehicle. Recombinant vaccine constructs can be expressed in the cytoplasm, displayed at the cell surface, secreted in the surrounding environment, or delivered in outer membrane vesicles.

**Table 1 antibiotics-11-01701-t001:** *Helicobacter pylori* vaccine under the preclinical and clinical phases of development.

Vaccine	Country	Status	Type	Route	Refs.
Recombinant UreB/LTB fusion	China	Phase III	Prophylacitc/Therapeutic	Oral	[24]
Imevax/IMX101 (*H. pylori* GGT)	Germany	Phase I	Prophylacitc/Therapeutic	Intradermal and sublingual	[25]
HelicoVax (HLA class II epitopes)	USA	Preclinical	Prophylactic	Intramuscular and intranasal	[26]
Recombinant CTB-UreI-UreB	China	Preclinical	Prophylactic	N/A	[27]
Recombinant *Vibrio cholerae* expressing *H. pylori* HpaA antigen	Sweden	Preclinical	Prophylactic	N/A	[28]
CTB-Lpp20	China	Preclinical	Prophylactic/therapeutic	Intraperitoneal	[29]
Recombinant HtrA	Australia	Preclinical	Prophylactic	N/A	[30]
Recombinant VacA-CagA-NAP	Germany	Phase I/II	Prophylactic	Intramuscular	[31]
Attenuated *Shigella* expressing UreB-HspA fusion	China	Preclinical	Prophylactic	Oral	[32]
Recombinant *Salmonella* expressing Urease or Urease + CT fusion	Germany	Preclinical	Prophylactic	Oral	[33]
Multi epitope: NAP, Urease, HSP60, and HpaA	China	Preclinical	Profilacitc/Therapeutic	Oral	[34]

NA: not applicable/available; CagA, cytotoxin-associated gene A; CTB: cholera toxin B subunit; GGT, gamma-glutamy transpeptidase; HpaA, *Helicobacter pylori* adhesin A; HSP, heat shock protein; HtrA, high-temperature requirement A protease; LTB, heat-labile enterotoxin B subunit; NAP, neutrophil-activating protein; VacA, vacuolating cytotoxin A, UreB, urease B subunit; UreI, urease I subunit.

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
