# Peer review of "Recombinant Helicobacter pylori Vaccine Delivery Vehicle: A Promising Tool to Treat Infections and Combat Antimicrobial Resistance"

_antibiotics, 2022, doi:10.3390/antibiotics11121701_

Round 1

Reviewer 1 Report

The manuscript entitled " Vaccination to tackle antimicrobial resistance: the potential of recombinant Helicobacter pylori vaccine delivery vehicle " contains an interesting topic. The authors have performed a review of the use of antibiotic-based treatment for Hp infection and the potential of vaccine delivery vehicle by Hp. This paper summarizes the research progress of AMR of Hp and the Hp delivery platform.

1. The second part of the paper, “Antimicrobial resistance and management of Hp”, should be deleted. The review should focus on the research of Hp vaccine delivery platform. AMR of Hp is another topic.

2. The review summary is not comprehensive enough, there are omissions. such as, in “4.3. Stabilization of recombinant plasmids in Helicobacter pylori” the author only introduces the balanced-lethal system, antibiotic-free plasmid stabilization systems, developed in salmonella. There are many other systems that have not been covered, such as the plasmids of Lactococcus lactis.

3.The picture of the paper is too simple and rough, please make it again.

4.The list of references lacks many of the up-to-date literature.

Author Response

Responses to Reviewer

  1. The second part of the paper, “Antimicrobial resistance and management of Hp”, should be deleted. The review should focus on the research of Hp vaccine delivery platform. AMR of Hp is another topic.

We thank the reviewer for this important suggestion that calls for clarification. This review is destined to a special issue of Antibiotics dedicated to antimicrobial resistance (AMR) and H. pylori. As you know, vaccination is an important strategy for addressing the raise of AMR that is advocated by global organizations like WHO, FAO and OIE that are at the forefront of this effort. Since (i) AMR in H. pylori is especially concerning in low-income countries for reasons discussed in the paper (Section 2, lines 82-90 of the initial manuscript; lines 86-94 of the revised version) and (ii) development of vaccines for H. pylori has been unsuccessful so far, we intended to focus our manuscript to a vaccine strategy never explored in this bacterium and that could be an ideal means for addressing the problems posed by AMR in H. pylori, especially in low-income countries. For this reason, we felt that a short summary on AMR in H. pylori is relevant in this review. However, to go toward the reviewer’s suggestion and to better illustrate the link between AMR and the vaccine strategy we propose, we added a sentence in the introduction, lines 44-46 and modified the title of Section 2 (line 64) of the revised manuscript.

  1. The review summary is not comprehensive enough, there are omissions. such as, in “4.3. Stabilization of recombinant plasmids in Helicobacter pylori” the author only introduces the balanced-lethal system, antibiotic-free plasmid stabilization systems, developed in Salmonella. There are many other systems that have not been covered, such as the plasmids of Lactococcus lactis.

The idea of using recombinant bacteria as vaccine delivery vehicle has been investigated in several species but, to our knowledge, there are not available references of such work in H. pylori. The essence of this review is to show that existing technologies enable the construction of recombinant H. pylori that present advantages in serving as vaccine delivery vehicles. Our goal was not to review the existing technologies but rather to discuss how examples of the existing technologies can be implemented in H. pylori.

Regarding section 4.3, we agree with the reviewer in that approaches other than the balanced-lethal systems developed should at least be mentioned. We change the title of this section and substantially modified its content mentioning other system and their corresponding references. We kept the balance-lethal system of Salmonella as a good example and explain the reason for this: Salmonella and Hp belong to the same category of bacterial delivery vehicle (attenuated pathogen) and the homologs of the genes involved in this system exist in Hp.

  1. The picture of the paper is too simple and rough, please make it again.

We assume that the reviewer would like to mention Figure 3 in this comment. We do agree that this figure is simple. Our intention was just to show the four vaccine delivery routes that can be considered. We suggest keeping this figure and if not suitable, we will agree to simply remove it.

  1. The list of references lacks many of the up-to-date literature

We have updated the list of references mentioned recent publications elated to different sections of the review.

We would like to thank the reviewer for his constructive comments that contributed to improving the quality of our manuscript.

Reviewer 2 Report

The article summarizes the published studies on the use of antibiotic-based treatment for Helicobacter pylori (Hp) infection and its related antimicrobial resistance problems together with a brief description of the status of vaccine development for Hp. It's a well written manuscript. However, I believe following addition/editing will improve the manuscript quality and will be beneficial for the reader.

1. The article mainly focus on Hp vaccine, but the first part of title "Vaccination to tackle antimicrobial resistance: ..." sounds more of a broader coverage of overall antimicrobial resistance issue. So, the title can be modified.

2. Article briefly glossed over the status of vaccine development for Hp, a table listing all the vaccines under development with their development stage will be great.

3. In segment 4.6, not sure what authors meant by "Overcoming oral tolerance" should it be overcoming the stability issues for oral administration. Authors are requested to clarify.

 4. Please elaborate on Biocontaintment  and strategies?

Author Response

Responses to Reviewer

  1. The article mainly focuses on Hp vaccine, but the first part of title "Vaccination to tackle antimicrobial resistance: ..." sounds more of a broader coverage of overall antimicrobial resistance issue. So, the title can be modified.

We thank the reviewer for this suggestion. We changed the title of the manuscript to “Recombinant Helicobacter pylori vaccine delivery vehicle: a promising tool to treat infections and combat antimicrobial resistance”

  1. Article briefly glossed over the status of vaccine development for Hp, a table listing all the vaccines under development with their development stage will be great.

A table listing the vaccines being developed against H. pylori was added as Table 1 of the revised manuscript.

  1. In segment 4.6, not sure what authors meant by "Overcoming oral tolerance" should it be overcoming the stability issues for oral administration. Authors are requested to clarify.

We thank the reviewer for bringing this to our attention. Oral tolerance is a specific immune unresponsiveness to an antigen delivered by the oral route. It serves to prevent untimely activation of the immune system at the gastrointestinal mucosa when there are lots of non-self molecules. We have added a sentence line 321 of the revised version and added as a reference a recent review on oral tolerance to take into account the reviewer’s remark.

  1. Please elaborate on Biocontainment and strategies?

We thank the reviewer for this important suggestion. We have added content Section 4.7 that elaborate further on this topic.

We would like to thank the reviewer for his positive comments and for the interesting suggestions and guidance he provided.

Round 2

Reviewer 1 Report

I have no question.